# Aminomethanesulfonic acid illuminates the boundary between full and partial agonists of the pentameric glycine receptor

**Josip Ivica[1†‡], Hongtao Zhu[2,3†], Remigijus Lape[1‡], Eric Gouaux[2,4*], Lucia G Sivilotti[1*]**

[1]Department of Neuroscience, Physiology and Pharmacology, Division of Biosciences, University College London, London, United Kingdom; [2]Vollum Institute, Oregon Health and Science University, Portland, United States; [3]Laboratory of Soft Matter Physics, Institute of Physics, Chinese Academy of Sciences, Beijing, China; [4]Howard Hughes Medical Institute, Oregon Health & Science University, Portland, United States

**Abstract** To clarify the determinants of agonist efficacy in pentameric ligand-gated ion channels, we examined a new compound, aminomethanesulfonic acid (AMS), a molecule intermediate in structure between glycine and taurine. Despite wide availability, to date there are no reports of AMS action on glycine receptors, perhaps because AMS is unstable at physiological pH. Here, we show that at pH 5, AMS is an efficacious agonist, eliciting in zebrafish α1 glycine receptors a maximum single-channel open probability of 0.85, much greater than that of β-alanine (0.54) or taurine (0.12), and second only to that of glycine itself (0.96). Thermodynamic cycle analysis of the efficacy of these closely related agonists shows supra-additive interaction between changes in the length of the agonist molecule and the size of the anionic moiety. Single particle cryo-electron microscopy structures of AMS-bound glycine receptors show that the AMS-bound agonist pocket is as compact as with glycine, and three-dimensional classification demonstrates that the channel populates the open and the desensitized states, like glycine, but not the closed intermediate state associated with the weaker partial agonists, β-alanine and taurine. Because AMS is on the cusp between full and partial agonists, it provides a new tool to help us understand agonist action in the pentameric superfamily of ligand-gated ion channels.

## Editor's evaluation

Ivica et al. provide both functional and structural characterization of a relatively unstudied glycine receptor agonist. Their work supports their prior conclusions regarding the function of full vs. partial agonists, and provides a new look at a ligand that is structurally in between a full and partial agonist. This manuscript will be of interest to both biophysical and pharmacological investigations of ligand-gated ion channels.

## Introduction

The glycine receptor (GlyR), a member of the pentameric ligand-gated ion channel superfamily, is an anion-permeable channel that mediates fast synaptic inhibition in caudal areas of the central nervous system, particularly in the spinal cord. Studies of the GlyR have been instrumental in illuminating structure-function relationships and the molecular activation mechanism in the pentameric

**\*For correspondence:**
gouauxe@ohsu.edu (EG);
l.sivilotti@ucl.ac.uk (LGS)

[†]These authors contributed equally to this work

**Present address:** [‡]Laboratory for Molecular Biology, Cambridge, United Kingdom

**Competing interest:** The authors declare that no competing interests exist.

**Figure 1.** AMS is a highly efficacious agonist on zebrafish α1 GlyR. (**A**) Structures of glycine, β-alanine, AMS, and taurine. (**B**) Whole-cell current responses of HEK293 cells elicited by application of agonist solutions (pH 5) with a U-tube. Cells were held at −40 mV. (**C**) Average concentration-response curves for glycine (black), β-alanine (green), AMS (blue), and taurine (red), n=6–9 cells. Responses of AMS, β-alanine, and taurine are normalized to those to a saturating concentration of glycine (100 mM) in each cell. AMS, aminomethanesulfonic acid.

The online version of this article includes the following source data for figure 1:

**Source data 1.** Data for the pooled dose-response curves in the figure.

superfamily of receptors, because GlyR is well-suited both to single-channel recording to quantify detailed activation mechanisms (*Burzomato et al., 2004*; *Lape et al., 2008*) and to high-resolution structural investigations (*Du et al., 2015*).

We have recently shown that the degree of contraction induced by the agonist binding in the orthosteric neurotransmitter site is important in determining the efficacy with which agonists open this channel (*Yu et al., 2021*). Thus, the smallest agonist, the natural transmitter glycine, is the most efficacious agonist and when the channel is fully occupied by glycine, only open and desensitized structures can be detected. In single-channel recordings, a glycine-bound channel is either desensitized or open for more than 95% of the time and we shall refer to glycine as a full agonist. Larger compounds are weaker as agonists (e.g., partial agonists), and in their presence a third structural state is seen, an intermediate state where the binding site has closed on the agonist, but the pore is still in the resting closed conformation. A particularly well-characterized partial agonist is taurine, which produces approximately half of the maximum open probability response seen with glycine. Taurine (*Figure 1A*) has a bulkier anionic moiety than glycine (sulfonate instead of carboxylate) and has one additional methylene group separating the amino and anionic entities, respectively. As we inspected the structure of agonist-occupied GlyR binding sites, we wondered which of these two features was most important for determining agonist efficacy. We already knew that β-alanine, the carboxylate homolog of taurine, is less efficacious than glycine, but more efficacious than taurine. This

strongly suggests that a structure intermediate between glycine and taurine, aminomethane sulfonic acid (AMS), should also be an agonist, and probably more efficacious than taurine. Surprisingly, we found only one study in the literature with data on AMS. In 1973, (*Young and Snyder, 1973*) reported that AMS displaced the binding of radioactive strychnine, a competitive antagonist of GlyR, but had no agonist effects when applied by iontophoresis onto native GlyRs in spinal cord neurons.

Here, we show that AMS is an efficacious GlyR agonist, whose activity was not detected in the past because it is chemically unstable at physiological pH and must be kept at acidic pH when employed in experimental studies. At acidic pH, GlyR gating is diminished, and taurine becomes a weaker agonist, whereas the maximum open probability elicited by glycine remains very high. Under these conditions, AMS is a strong agonist, more efficacious than β-alanine and almost as efficacious as glycine. Single particle cryo-electron microscopy (EM) structures of the AMS-bound GlyRs have features similar to those we reported for the GlyR bound to the most efficacious of the agonists, glycine, in that they populate only the open and the desensitized states, and have a compact agonist binding pocket. While the instability of AMS at neutral pH limits its general usefulness as an agonist, its efficacy provides a new tool to test hypotheses for the structural correlates of agonist action in pentameric ligand-gated channels.

## Results

In the initial experiments, we dissolved AMS in a pH 7.4 solution and tested by whole-cell recording its effect at 100 mM on α1 GlyRs expressed in HEK 293 cells. Currents elicited by AMS were found to be inconsistent in amplitude over time. We also found that the pH of AMS solutions was unstable, drifting in a matter of few minutes. We hypothesized that this drift reflected AMS instability and decomposition at neutral pH. We then tried dissolving the compound in acidic solutions and found that at pH 5, 100 mM AMS solutions were stable, and remained within 0.1 of a unit of the initial pH for almost an hour, enough time to test their effect on GlyRs.

### AMS is a highly efficacious GlyR agonist at acidic pH

*Figure 1B* shows whole-cell recordings of the responses of zebrafish α1 GlyR expressed in HEK 293 cells to U-tube applications of glycine, β-alanine, AMS, or taurine at pH 5. In order to maintain stable recordings, the cells were kept at physiological extracellular pH and only the agonist solutions were at pH 5. AMS had a strong agonist effect and evoked currents similar in amplitude and time course to those produced by glycine. Normalizing AMS responses against the maximum response to glycine in the same cell showed that AMS was almost as efficacious as glycine (89%; *Figure 1C*, *Table 1*). The other agonists, β-alanine and taurine, were clearly partial agonists, eliciting 69% and 18% of the maximum glycine response, respectively (*Figure 1C*, *Table 1*). Glycine is the most efficacious agonist known for this channel and was also the most potent agonist, with an $EC_{50}$ of 0.98 mM, followed by β-alanine (4.5 mM), whereas taurine and AMS had similar low potencies (7.9 and 8.7 mM, respectively; *Table 1*).

We have recently shown (*Ivica et al., 2022*) that even modest extracellular acidification (pH 6.4) reduces both the potency and the efficacy of agonists on GlyR. This is confirmed at pH 5, where glycine, β-alanine, and taurine have lower potency than at physiological pH (*Table 1*). The $EC_{50}$ values of glycine, β-alanine, and taurine increased from 0.19, 0.3, and 1.08 mM, their values at pH 7.4, to 0.98, 4.5, and 7.87 mM, respectively, at pH 5. At acidic pH, there was also a decrease in the maximum

**Table 1.** Whole-cell parameters for the action of agonists on the zebrafish α1 GlyR at pH 5.

| | $I_{max}$, nA | $EC_{50}$, μM | nH | $I_{agonist}/I_{Glymax}$ | n |
|---|---|---|---|---|---|
| Glycine | 4.3±1.3 | 980±360 | 1.20±0.21 | 1 | 8 |
| AMS | 5.8±1.8 | 8700±3100 | 1.95±0.22 | 0.89±0.06 | 9 |
| β-alanine | 3.5±0.3 | 4500±2600 | 1.20±0.40 | 0.69±0.09 | 6 |
| Taurine | 1.1±0.5 | 7900±2800 | 0.85±0.15 | 0.18±0.08 | 6 |

The online version of this article includes the following source data for table 1:

**Source data 1.** Raw data for the dose-response curves.

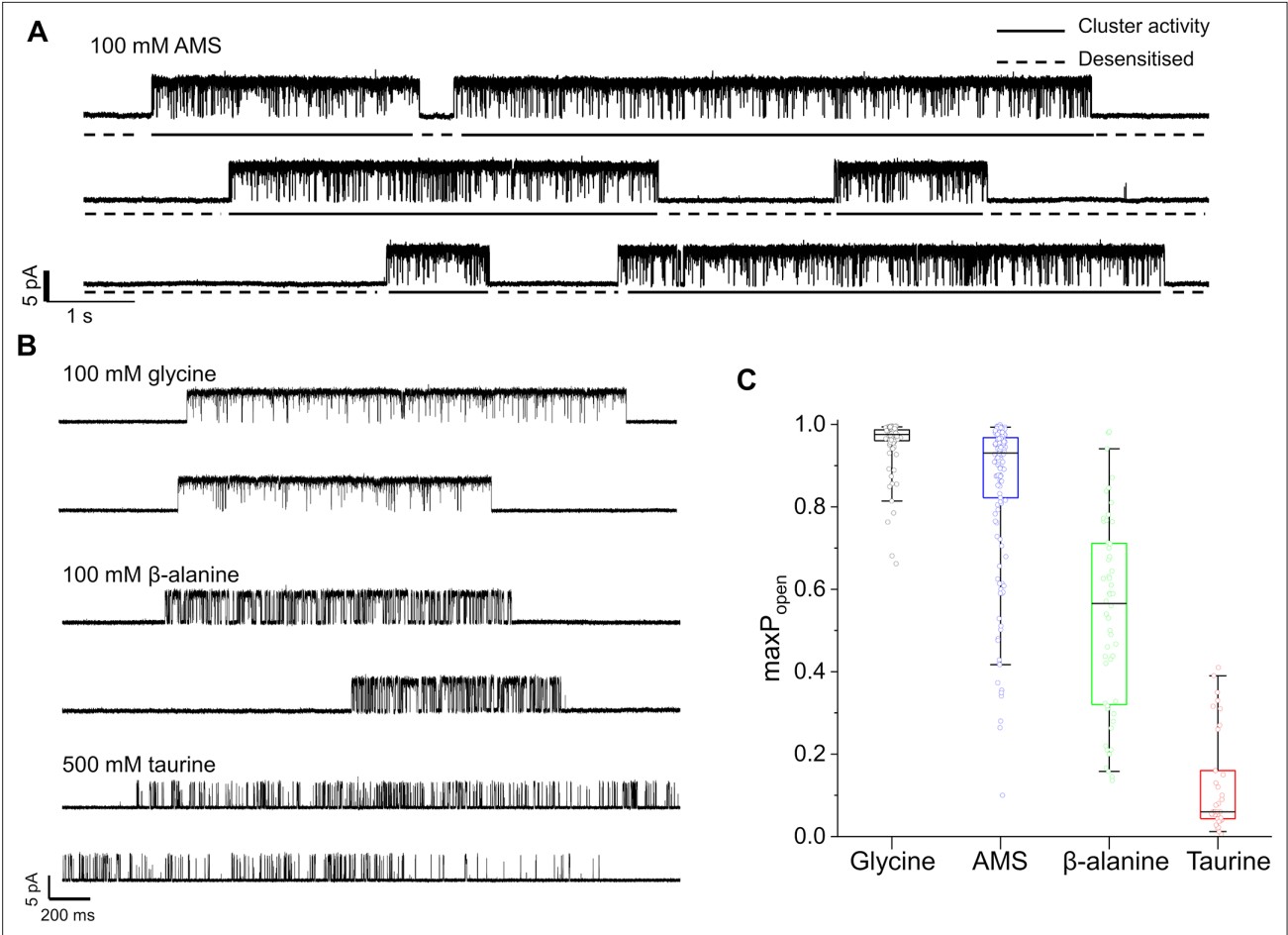

**Figure 2.** Maximum open probability evoked by different GlyR agonists. (**A, B**) Representative single-channel current recordings of zebrafish α1 GlyR activity evoked by high concentrations of agonists. Recordings were made in the cell attached configuration at +100 mV holding potential. (**C**) Boxplots of maximum Popen values for clusters with the different agonists (one point per cluster). Boxes and whiskers show the 25th and 75th and the 5th and 95th percentiles, respectively. The horizontal black line in the box is the median.

The online version of this article includes the following source data for figure 2:

**Source data 1.** Raw data for the agonist maximum Popen.

responses of β-alanine and taurine, to 69% and 18% of the maximum glycine responses (cf. 84 and 40% at physiological pH, respectively).

## Single-channel recordings

The whole-cell recordings showed that AMS is almost as efficacious as glycine on GlyR, but whole-cell data cannot give an absolute measurement of agonist efficacy. Given the impairment in gating at acidic pH, we do not know how efficacious glycine is at pH 5.

We therefore measured the single-channel maximum open probability (*Popen*) elicited by high concentrations of the four agonists at pH 5.

*Figure 2A* shows continuous cell-attached recordings in the presence of 100 mM AMS in the pipette at pH 5. 'Clusters' of openings are separated by long closed intervals that are the expression of desensitization. The high *Popen* of the AMS clusters confirms that this compound is a highly efficacious agonist on GlyR. The similar traces below (*Figure 2B*) show that glycine is still very efficacious at pH 5. Despite the acidic pH, glycine clusters have a high *Popen*, somewhat higher than the AMS clusters. This impression is confirmed by the boxplots of *Figure 2C*, where the *Popen* values of 146 AMS clusters (from 11 patches) have a mean of 0.85, close to the 0.96 value of glycine (*Table 2*; p<<0.001, two-tailed randomization test).

**Table 2.** Single-channel parameters of responses elicited by four agonists on zebrafish α1 GlyR. maxPopen was measured from n clusters of activation and reported as mean ± SD. Data at pH 7.4 are from *Ivica et al., 2021*.

| pH | | Glycine | β-alanine | AMS | Taurine |
|---|---|---|---|---|---|
| 5 | maxP$_{open}$ | 0.96±0.06 | 0.54±0.24 | 0.85±0.19 | 0.12±0.12 |
| | median P$_{open}$ | 0.976 | 0.566 | 0.931 | 0.060 |
| | n$_{patches}$(n$_{clusters}$) | 8 (92) | 7 (52) | 11 (146) | 9 (37) |
| | Agonist concentration (mM) | 100 | 100 | 100 | 500 |
| 7.4 | maxP$_{open}$ | 0.97±0.05 | 0.91±0.21 | / | 0.66±0.24 |
| | median P$_{open}$ | 0.989 | 0.978 | / | 0.728 |
| | n$_{patches}$(n$_{clusters}$) | 10 (48) | 7 (30) | / | 7 (71) |
| | Agonist concentration (mM) | 10 | 30 | | 100 |

The gating inhibition of GlyR at acidic pH was clear for the other two agonists. The compound β-alanine is an efficacious agonist on zebrafish α1 GlyR at pH 7.4 (maximum *Popen*=0.91; *Ivica et al., 2021*), but its maximum *Popen* is only 0.54 at pH 5 (*Figure 2B and C*), where it is clearly less efficacious than glycine and AMS.

Among the four agonists, taurine was the least efficacious. The mean maximum *Popen* measured from clusters of activation evoked by 500 mM taurine was 0.12±0.12, a value five times smaller than at pH 7.4 (0.66; *Ivica et al., 2021*).

## Cryo-EM structure determination of AMS-bound GlyR

We used styrene maleic acid (SMA) polymer to extract recombinant, zebrafish α1 GlyRs, together with endogenous lipids (*Figure 3—figure supplement 1*), because our earlier work showed preservation of physiological receptor states with this reagent (*Yu et al., 2021*). We found that while AMS triggered the aggregation of GlyRs, we could minimize receptor self-association by employing continuous thin carbon film as a support. Because of the instability of AMS at neutral pH (see Discussion), the GlyR cryo-EM grids were flash-frozen in less than 1 min after adding the ligand (see Materials and methods for details).

The single-channel data show that AMS is highly efficacious and produced clusters of openings (*Figure 2A*) that lack the long shut states seen with partial agonists and have a high maximum *Popen*, approaching that of glycine. We therefore hypothesized that AMS-bound GlyR should populate only open and desensitized states. In agreement with our hypothesis, the single particle cryo-EM analysis revealed open, desensitized and expanded-open states (*Figure 3*, *Supplementary file 1*, *Figure 3—figure supplement 2*), similar to our findings with glycine (*Yu et al., 2021*). Like with glycine, with AMS we did not capture the agonist-bound (intermediate) closed state seen with the partial agonists taurine and GABA (*Yu et al., 2021*). The overall resolutions for open, desensitized, and expanded-open states was 2.8 Å, 2.9 Å, and 3.1 Å, respectively. Importantly, these reconstructions have well-resolved extracellular domain (ECD) and transmembrane domain (TMD) densities, allowing us to observe conformational differences (*Figure 3—figure supplement 3*, *Supplementary file 1*) in the structures.

The pore domains of AMS-bound states adopt conformations similar to those we reported for glycine-bound and taurine-bound states. The tilted conformation of the M2 helices in both the desensitized and open state creates a constriction of the pore at the Pro residue in the –2′ position (*Figure 4A–B*). For the AMS-bound desensitized state, the diameter of this constriction is 3.2 Å, too narrow to allow the permeation of partially hydrated chloride ions (*Bormann et al., 1987*; *Hille, 2001*), confirming this state is non-conducting (*Figure 4A*). For the AMS-bound open state, the constriction of the pore is 5.4 Å in diameter (*Figure 4B*), indicative of a conducting state (*Yu et al., 2021*). The pore radius plots illustrate the similarities between the structures of the receptor bound to different agonists in the open and desensitized states (*Figure 4C–D*).

The neurotransmitter binding site is the first element of the receptor to productively interact with agonists and examining conformational changes in this area helps us understand agonist-induced

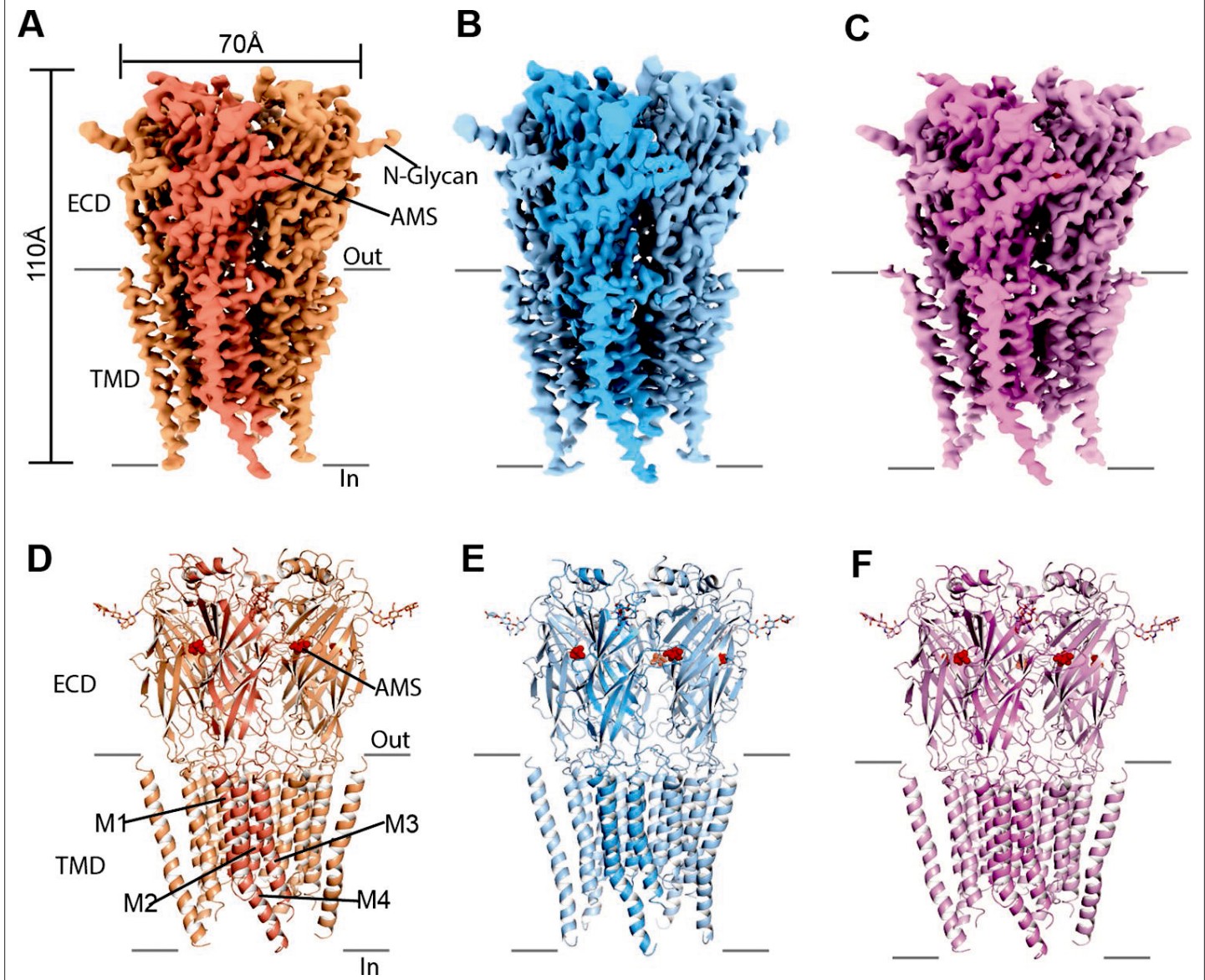

**Figure 3.** Cryo-EM analysis of structures of zebrafish α1 GlyR bound to AMS. (**A–C**) Cryo-EM density maps for desensitized, open, and expanded-open states with one subunit highlighted. The AMS density is in red. (**D–F**) Atomic models for desensitized, open, and expanded-open states. Shown are GlyR in cartoon representation, AMS in sphere representation (red), and *N*-glycans in stick representation. AMS, aminomethanesulfonic acid; EM, electron microscopy.

The online version of this article includes the following source data and figure supplement(s) for figure 3:

**Figure supplement 1.** GlyR purification.

**Figure supplement 1—source data 1.** Raw data for the original gels after SEC.

**Figure supplement 2.** Flow chart for cryo-EM data processing of GlyR bound with AMS.

**Figure supplement 3.** 3D reconstruction of GlyR-AMS states.

channel activation. Guided by the AMS density (*Figure 5—figure supplement 1A*), AMS can be unambiguously placed in the binding pockets. In both the desensitized and open states, the densities contributed by AMS are well resolved, with the larger sulfonate group at the entrance of the binding pockets (*Figure 5—figure supplement 1A*). All agonist binding sites appear fully occupied by AMS molecules, confirming that the compound had not degraded in our experimental conditions. Like with the full agonist glycine (*Du et al., 2015*; *Yu et al., 2021*), the amino group of AMS is sandwiched by residues on the (+) subunit, loop B F175 and loop C F223, with distances compatible with a cation-π

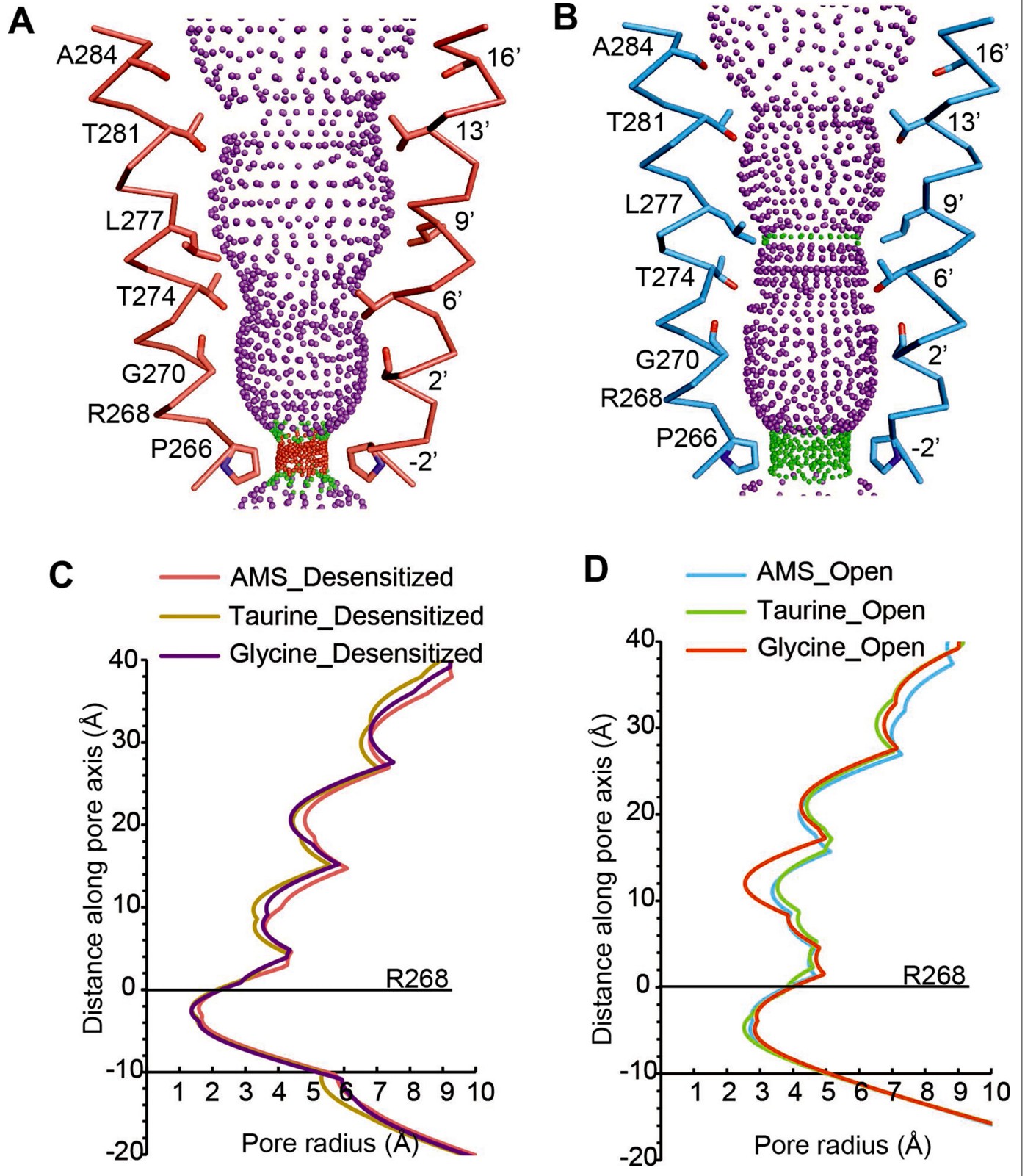

**Figure 4.** Comparison of ion channel pores. (**A, B**) Shape and ion permeation pathway for AMS-bound desensitized (see also (**C**)) and open (see also (**D**)) states. M2 helices and key amino acids are shown in ribbon and stick representation, respectively. Purple, green, and red spheres define radii >3.5 Å, 1.8–3.5 Å, and <1.8 Å. (**C, D**) Profiles of pore radii calculated by the HOLE program for desensitized (**A**) and open (**B**) states bound with AMS, taurine, and glycine. The Cα position of R268 was set to 0. AMS, aminomethanesulfonic acid.

interaction. These residues are important in GlyR agonist recognition (*Schmieden et al., 1993*) and form part of the canonical aromatic box of the binding site (*Pless et al., 2008*, reviewed in *Lynagh and Pless, 2014*). We observed also a potential hydrogen bond between the main chain carbonyl of loop B F175 and the amino group of AMS. At the other end of the agonist molecule, the sulfonate group participates in multiple hydrogen bonds with amino acids that include R81 and S145 derived from β strands 2 and 6 on the (−) subunit, respectively, and T220 in loop C of the (+) subunit (*Figure 5A–B*), all residues that are important for the agonist activation of GlyRs (*Grudzinska et al., 2005*; *Yu et al., 2014*; *Vandenberg et al., 1992*). The interactions we observed for AMS are similar to those of glycine (*Figure 5B–C*), underscoring the functional similarity between these two efficacious compounds.

Interestingly, we found that the amino group of the partial agonist taurine forms a hydrogen bond with loop B S174 (*Figure 5—figure supplement 1B*), a (+) side interaction that is absent in AMS and glycine, because of their shorter length. We had previously noted a similar possible interaction between loop B S174 and the amino group of another partial agonist, GABA (*Yu et al., 2021*). Considering that the interactions with the (−) subunit are similar for the two groups of agonists, full and partial, it is tempting to speculate that the difference in the amino group interactions on the principal side may contribute to the difference in their efficacy.

Our previous structural studies showed that agonist efficacy is correlated with the degree of contraction of the binding pocket (*Yu et al., 2021*). Because AMS is nearly as efficacious as glycine (*Figures 1 and 2*), we expect it to produce a contraction of the binding pocket similar to that caused by glycine. The volumes of glycine, AMS, and taurine binding pockets are 130 $Å^3$, 125 $Å^3$, and 151 $Å^3$, respectively, showing that the binding pocket, when bound with AMS or glycine, takes up a conformation that is more compact than that seen with the partial agonist taurine. By overlapping the binding pockets, we found that, in the glycine-bound site, the movement of loops B and C brings them closer to the ECD-TMD interface than in the AMS- and taurine-bound sites (*Figure 5D–E*, *Figure 5—figure supplement 1C*), which may be one reason for the higher efficacy of glycine.

The perturbation introduced by the agonist in the binding site spreads to the pore by eliciting conformational changes at the ECD-TMD interface, which can be detected when we compare the apo and open states (*Yu et al., 2021*). We scrutinized the conformational differences of the ECD-TMD in GlyR open states bound to different ligands. We found that the ECD-TMD interfaces are overall similar between glycine, AMS, and taurine (*Figure 6A–B*), as indicated by distances between the centers of mass of the secondary structure elements.

## Discussion

### Instability of AMS at physiological pH

Our initial findings, that AMS produced inconsistent responses, and that the pH of its solutions drifted relatively quickly from neutral pH, suggested chemical instability. Indeed, prior studies have shown that AMS is unstable and decomposes to formaldehyde and sulfur dioxide (*Frankel and Moses, 1960*; *Moe et al., 1981*). The mechanism of decomposition of AMS and other α amino sulfonic acids is thought to require the availability of the electron pair of the unprotonated amino group (*Moe et al., 1981*). This implies that AMS should be maximally stable when the amino group is protonated and the compound is a zwitterion. The amino group on AMS is not very basic, with a pKa value of 5.75 (cf. 9.06 and 9.78 for the amino groups of taurine and glycine, respectively; *Benoit et al., 1988*). At physiological pH, only 2% of AMS is expected to be in the stable zwitterion form. Full protonation requires pH values too acidic to be compatible with stable electrophysiological recordings. We opted to test AMS at a compromise pH of 5, where 85% of this compound is a zwitterion. In order to ensure cell health, we kept cells at neutral pH and switched them to pH 5 only during agonist application. In the structural studies, particular care was taken to freeze the grids as quickly as possible after mixing agonists and receptors and our analysis showed density features consistent with undegraded AMS molecules. We were rewarded in these experimental choices by our demonstration that AMS is an efficacious agonist at both functional and structural levels.

### AMS behaves structurally as a highly efficacious agonist

Our whole-cell and single-channel recordings demonstrated AMS is an efficacious agonist, which produces a mean maximum $P_{open}$ of 0.85 (cf. 0.96 for glycine) and is clearly more efficacious than

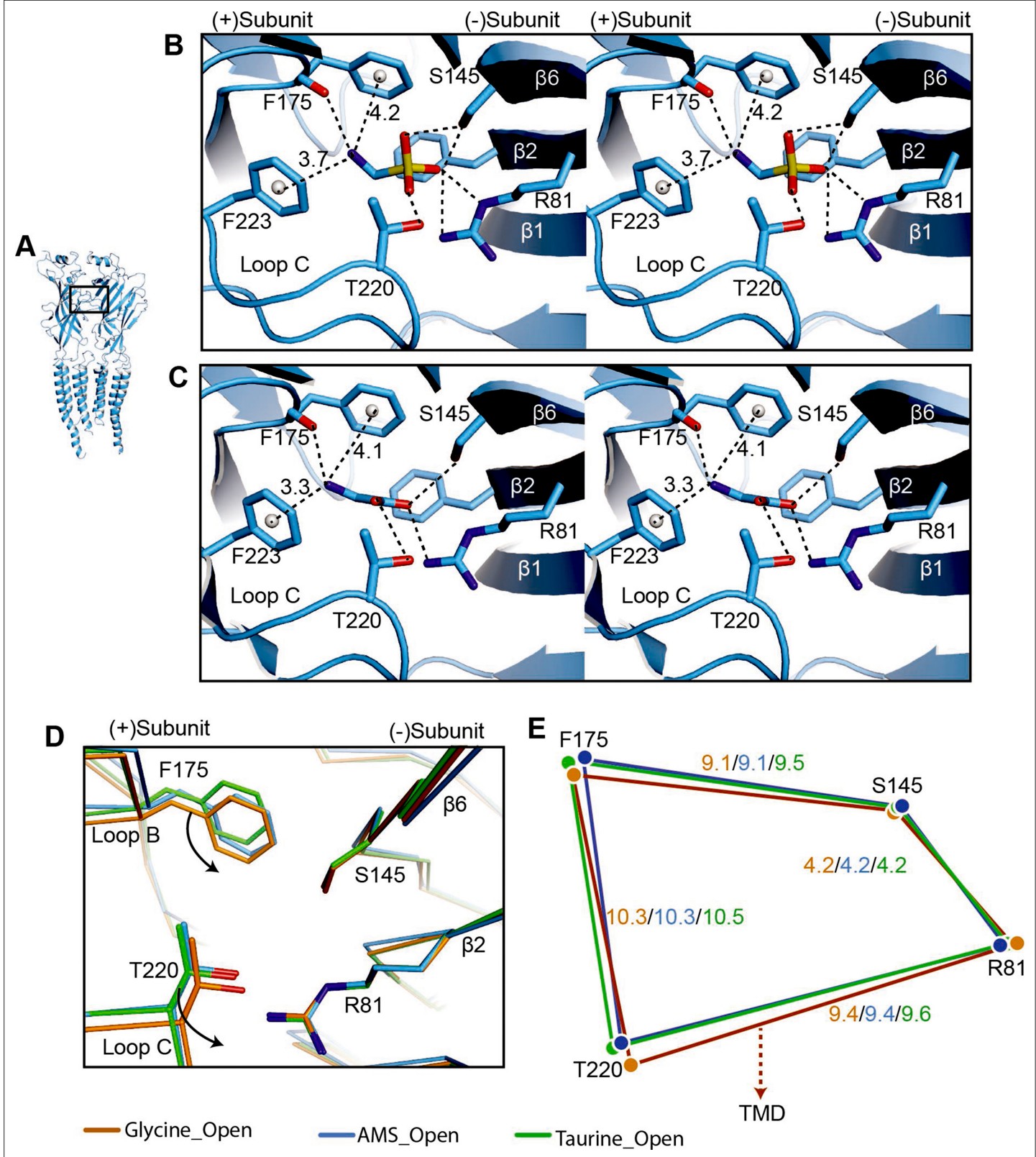

**Figure 5.** Comparison of agonist binding sites. (**A**) Two adjacent GlyR subunits are shown in cartoon representation. The agonist binding pocket is indicated by a black box. (**B, C**) Stereo figures of the binding sites showing likely hydrogen and cation-π interactions with AMS (**B**) and glycine (**C**) bound, respectively. Numbers are the distances in Å of probable cation- π interactions. Numbering of residues includes the signal peptide of 16 amino acids. (**D**) Comparison of the positions of key binding residues in the open states of the glycine (salmon), taurine (green), and AMS (blue) complexes, obtained

*Figure 5 continued on next page*

*Figure 5 continued*

by superposing the respective ECDs. (**E**) Schematic diagram illustrating the distances (Å) between the Cα atoms of key amino acids in glycine-, taurine-, and AMS-bound open states. AMS, aminomethanesulfonic acid; ECD, extracellular domain.

The online version of this article includes the following figure supplement(s) for figure 5:

**Figure supplement 1.** Comparison of loops B and C.

β-alanine, which until now was considered the second most efficacious agonist of GlyRs (***Ivica et al., 2021***).

Cryo-EM structural analysis of AMS-bound GlyRs detected all the structural features that we have associated with the high efficacy of glycine. The partial agonist-bound closed state (***Yu et al., 2021***) is absent from both the AMS-bound and glycine-bound GlyRs. Indeed, our data show that, for the AMS-bound receptor, 94.3% and 3.4% of the particles are in open and desensitized state classes, respectively (***Supplementary file 1***), consistent with the notion that the particle fractions for desensitized states are much smaller than open states when the receptor is isolated via SMA, compared to when it is incorporated into nanodiscs (***Yu et al., 2021***). Further confirmation comes from inspecting the agonist binding site. The most obvious feature is that the binding pocket is as compact when bound to AMS as with glycine (***Yu et al., 2021***). The measurements for the glycine-bound and AMS-bound pockets are almost identical, and both are more compact than those obtained with the partial agonist taurine. Interestingly, the potency of AMS is low, as its $EC_{50}$ is about tenfold higher than that of glycine, and similar to the $EC_{50}$ of taurine. Thus, even though AMS has an additional oxygen for the formation of another hydrogen bond, it must bind with much lower affinity than glycine, underscoring the lack of correlation between efficacy and affinity of these agonists.

Both AMS and glycine form multiple interactions with the (+) and (−) subunits (***Figure 5B***). These interactions are similar but not identical for partial and full agonists, and we were able to observe some differences. For instance, the interaction between loop B S174 and the agonist amino group, which was observed for partial agonist taurine (***Figure 5—figure supplement 1B***) and GABA (***Yu et al., 2021***), was not seen with either AMS and glycine (***Figure 5A–B***) probably because these molecules are shorter. It is tempting to speculate that this additional interaction may limit the movement of loop B, a domain whose conformation has a close relationship with the ECD-TMD interface. This limitation in movement may contribute to the low efficacy of taurine and GABA.

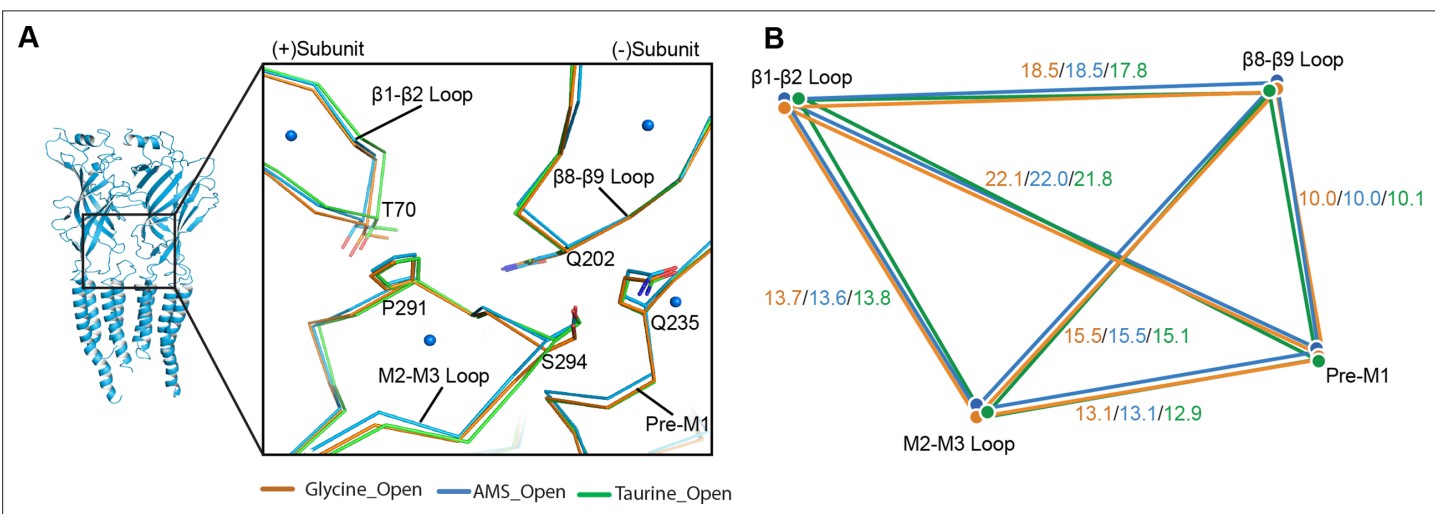

**Figure 6.** Comparison of the ECD-TMD interface in different agonist-bound complexes. (**A**) Superposition of the ECD-TMD interface of the open states of the glycine (salmon), taurine (green), or AMS (blue) bound forms. The key amino acids at the ECD-TMD interface are shown in stick representation. Key secondary structure elements are labelled. The blue spheres represent the centers of mass of the secondary structure elements for the AMS-bound structure. (**B**) Schematic diagram illustrating the distances (Å) of the center of mass points shown in panel (**A**) of glycine-, taurine-, and AMS-bound open states. ECD, extracellular domain; TMD, transmembrane domain.

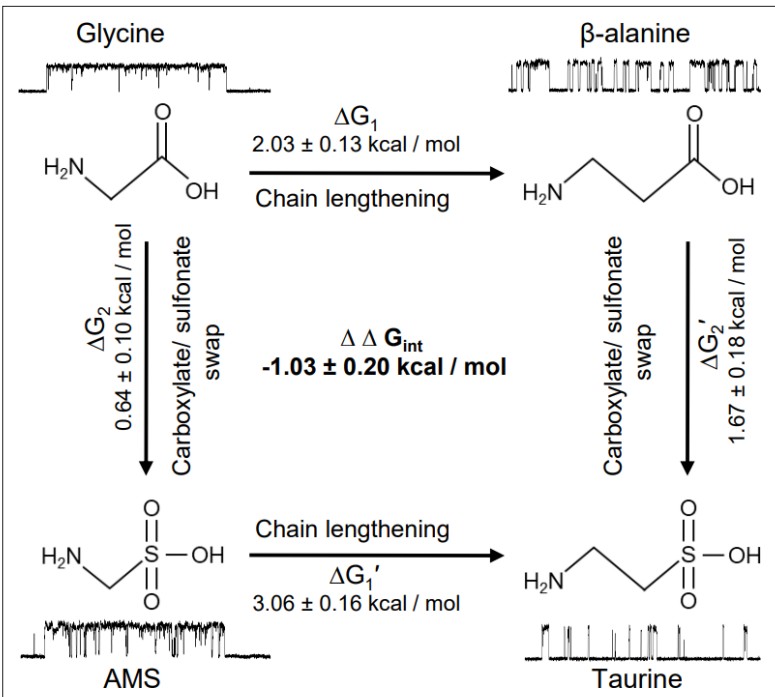

**Figure 7.** Thermodynamic cycle for the four GlyR agonists functionally characterized, showing their structure and example sweeps of the single-channel activity they elicit.

## Glycine has uniquely high efficacy

The natural agonist of GlyRs, the neurotransmitter glycine, remains the most efficacious agonist. It has generally been thought that this is simply due to glycine being the smallest molecule, but our discovery of the agonist effects of AMS allowed us to consider this question in greater detail, examining the effects of making the agonist longer or changing its anionic moiety from a carboxylate to a larger sulfonate group. In our functional work, we characterized a panel of four agonists, chosen on the basis of their related structures. *Figure 7* shows how swapping the carboxylate group with sulfonate (vertical arrows) in glycine and β-alanine produces AMS and taurine, respectively, whereas lengthening the carbon chain by one methylene (horizontal arrows) in glycine and AMS produces β-alanine and taurine, respectively.

*Figure 7* shows how both changes in structure decrease agonist efficacy. Of the two structure modifications we tested, lengthening the distance between the two charges on these amino acids causes the greater decrease in efficacy. The example clusters in *Figure 7* show that the median maximum open probability decreases from 0.98 to 0.57 (glycine to β-alanine, corresponding to a decrease in $E_{eff}$ from 40.7 to 1.30) and from 0.93 to 0.06 (AMS to taurine, corresponding to a decrease in $E_{eff}$ from 13.5 to 0.06). It is worth noting that further lengthening β-alanine by one methylene produces GABA, a weak partial agonist.

Increasing the size of the negatively charged group from carboxylate to sulfonate had a modest effect for the shorter agonists, and the overall efficacy of glycine was decreased by less than twofold with the change to AMS. The effect of this swap is greater on the longer agonists. This can better be examined by thermodynamic cycle analysis of the free energy changes associated with gating for the four related agonists (*Figure 7*). These values can be obtained from our estimates of overall efficacy yielded by single-channel measurements of maximum open probability. In the thermodynamic cycle analysis of the free energy values that underlie gating, we found that there is marked coupling between the effects of the two changes in agonist structure, with an estimated coupling energy of 1.03±0.2 kcal/mol (*Figure 7*). This means that the combined effect is more than we would anticipate by the linear sum of the effects of each change. In other words, taurine is less efficacious than we would predict. Conversely, glycine is more efficacious than we would expect it to be, given that it is a shorter taurine, where the sulfonate is replaced by a carboxylate.

Our previous work has shown that the much greater efficacy of glycine compared with taurine or GABA is associated with a tighter conformation of the binding pocket bound to glycine. Our new data show that AMS is almost as full an agonist as glycine and that the binding pocket bound to AMS is as compact as with glycine, at the resolution of our data. However, the structural correlates for the small difference in efficacy between AMS and glycine have so far proven elusive and may require further structural studies, at higher resolution.

In conclusion, our electrophysiological and cryo-EM experiments demonstrate that AMS is an agonist with high efficacy, greater than that of β-alanine and second only to that of glycine. The availability of AMS as a new tool is particularly useful, because structural changes with activation are relatively small in channels of this superfamily, and the differences between agonists of different efficacy are subtle. Our present work with AMS allowed us to confirm our previous observations with glycine on the structural correlates of agonist efficacy in GlyRs and also to propose new structural features that may be important in agonist activation of channels in this superfamily.

# Materials and methods

**Key resources table**

| Reagent type (species) or resource | Designation | Source or reference | Identifiers | Additional information |
|---|---|---|---|---|
| Cell line (*Homo sapiens*) | HEK293A | Invitrogen: https://www.thermofisher.com/order/catalog/product/R70507?SID=srch-hj-R705-07 | Catalog no. R705-07 | lot #1942007; certified mycoplasma free by supplier |
| Cell line (*Spodoptera frugiperda*) | Sf9 | Thermo Fisher Scientific: https://www.thermofisher.com/order/catalog/product/12659017 | Catalog no. 12659017 | lot 421973 mycoplasma tested at OHSU |
| Recombinant DNA reagent | Zebrafish GlyRa1 subcloned in pcDNA3 vectorf | https://doi.org/10.1016/j.cell.2021.01.026 | Uniprot identifier: O93430 | |
| Recombinant DNA reagent | Zebrafish GlyRa1 subcloned in pFastBac Dual vector | This paper | Uniprot identifier: O93430 | |
| Software, algorithm | OriginPro software | https://www.originlab.com/ | Origin 2019b | |
| Software | Clampex software, Molecular Devices | https://support.moleculardevices.com/s/article/Axon-pCLAMP-10-Electrophysiology-Data-Acquisition-Analysis-Software-Download-Page | Clampex 10.7 | |
| Software | DC-STATS | *Plested and Lape, 2020* | | Our lab; https://github.com/DCPROGS/DCSTATS/releases/tag/v.0.3.1-alpha |
| Software | CVFIT | *Lape, 2020b* | | https://github.com/DCPROGS/CVFIT/releases/tag/v1.0.0-alpha |
| Software | MotionCor2 | https://doi.org/10.1038/nmeth.4193 | RRID:SCR_016499 | http://msg.ucsf.edu/em/software/motioncor2.html |
| Software | cryoSparc | https://doi.org/10.1038/nmeth.4169 | RRID:SCR_016501 | https://cryosparc.com/ |
| Software | Coot | https://doi.org/10.1107/S0907444910007493 | RRID:SCR_014222 | https://www2.mrc-lmb.cam.ac.uk/personal/pemsley/coot/ |

*Continued on next page*

*Continued*

| Reagent type (species) or resource | Designation | Source or reference | Identifiers | Additional information |
|---|---|---|---|---|
| Software | Phenix | https://doi.org/10.1107/S2059798318006551 | RRID:SCR_014224 | https://www.phenix-online.org/ |
| Software | Pymol | PyMOL Molecular Graphics System, Schrodinger, LLC | RRID:SCR_000305 | http://www.pymol.org/ |
| Software | UCSF ChimeraX | https://doi.org/10.1002/pro.3235 | RRID:SCR_015872 | http://cgl.ucsf.edu/chimerax/ |
| Antibody | anti-human CD235a-APC (mouse monoclonal) | Thermo Fisher Scientific | Cat#: 17-9987-42; RRID:AB_2043823 | FACS (5 µl per test) |
| Recombinant DNA reagent | PLKO-GFP (plasmid) | This paper | | GFP version of pLKO.1-Puro |
| Recombinant DNA reagent | PLKO.1-Puro (plasmid) | Sigma-Aldrich | RRID:Addgene_10878 | Pol III based shRNA backbone |
| Sequence-based reagent | Gipc1_F | This paper | PCR primers | GGGAAAGGACAAAAGGAACCC |
| Sequence-based reagent | Gipc1_R | This paper | PCR primers | CAGGGCATTTGCACCCCATGCC |
| Sequence-based reagent | siRNA: nontargetin control | Thermo Fisher Scientific | 4390843 | Silencer Select |
| Peptide, recombinant protein | Streptavidin | Thermo Fisher Scientific | Cat#: 434302 | |
| Commercial assay or kit | In-Fusion HD Cloning | Clontech | Clontech: 639647 | |
| Chemical compound, drug | CBR-5884 | Sigma-Aldrich | SML1656 | |
| Chemical compound, drug | SL30010 (SMALP 30010P) | Polyscope | | http://polyscope.eu/markets/polyscience/ |

## Cell culture

HEK293A (from Life Technologies/Invitrogen, now Thermo Fisher Scientific; certified mycoplasma free by supplier) were maintained in an incubator at 37°C, 5% $CO_2$, and 95% humidity. The culture medium was Dulbecco's modified Eagle's medium (Gibco, 41966029) supplemented with 10% v/v heat-inactivated fetal bovine serum (Gibco), 100 units/ml penicillin G, and 100 µg/ml streptomycin sulfate (Gibco). Cell aliquots were frozen from the third passaging since the purchase batch. For experiments, an aliquot was thawed and then passaged up to 25 times, every 2–3 days after reaching 70–80% confluence.

For transfection, cells were plated on 13 mm poly-L-lysine glass coverslips, placed inside 35 mm cell culture dishes containing 2 ml growth medium. The transfection was performed with the calcium phosphate precipitation method (*Groot-Kormelink et al., 2002*). The total amount of DNA per coverslip was 3 µg. The DNA mixture contained 2% plasmid coding for zebrafish α1 GlyR (Uniprot accession number O93430), 20% plasmid coding for the enhanced green fluorescence protein (eGFP) and 78% of 'empty' pcDNA3 plasmid. The empty plasmid was introduced to optimize the level of receptor expression and the eGFP to identify the transfected cells. After 4–8 hr, the transfection medium was replaced with fresh growth medium. Electrophysiological recordings were performed 24–48 hr after transfection.

## Whole-cell recordings

Recording pipettes were pulled with a P-97 horizontal puller (Sutter Instruments), using thick-walled borosilicate capillaries (GC150F-7.5; Harvard Apparatus, UK). The tips were fire-polished with a micro-forger (Narishige, Japan) to a resistance of 3–5 MΩ. Currents were recorded with an Axopatch 200B

amplifier (Molecular Devices), prefiltered with the amplifier's built in 5 kHz low-pass Bessel filter and sampled at 20 kHz with Digidata 1550B digitizer (Molecular Devices) to a computer hard drive with Clampex 10.7 software (Molecular Devices). Currents were recorded at a nominal –40 mV holding potential (–50 mV if corrected for liquid junction potential). Access resistance was never higher than 8 MΩ and was compensated by 60–80%, with a maximum voltage error of 10 mV. For the figures, current traces were filtered with an additional 1 kHz low pass Gaussian filter in Clampfit 10.7.

The bath extracellular solution contained (in mM): 112.7 NaCl, 20 sodium gluconate, 2 KCl, 2 $CaCl_2$, 1.2 $MgCl_2$, 10 tetraethylammonium Cl (TEACl), 30 glucose, and 10 HEPES; the pH was adjusted with NaOH to 7.4.

Agonist solutions were freshly prepared in pH 5 extracellular solution. For this solution, the 10 mM HEPES buffer was replaced with 6.7 mM sodium acetate and 3.3 mM acetic acid.

The intracellular solution contained (in mM): 101.1 potassium gluconate, 11 EGTA, 6 KCl, 1 $CaCl_2$, 1 $MgCl_2$, 20 TEACl, 2 MgATP, 10 HEPES, and 40 sucrose; the pH was adjusted to 7.2 with KOH.

Agonists were applied to a cell with a custom-made U-tube. The speed of agonist application was determined by positioning the recording pipette just above the cell and measuring the 20–80% rise time of the current evoked by U-tube application of diluted bath solution (50%) to the recording pipette. Rise time was typically 2 ms (the U-tube tool was discarded if rise time was slower than 20 ms).

Agonist was applied every 30 s including application of a saturating concentration of 100 mM glycine every 3–4 applications to monitor current rundown. If the rundown was more than 30%, the cell was discarded from analysis. A full dose-response curve was obtained in each cell, and the responses measured with Clampfit 10.7 were fitted with the Hill equation with custom-made software (*Lape, 2020a*). The parameters of the dose response fits obtained from n cells ($EC_{50}$, Imax, and nH) are reported as mean ± SD. For the figures, the dose-response curves were normalized to the maximum response in each cell, pooled together and refitted with the Hill equation.

## Single-channel recordings

Glass pipettes were fabricated with a Sutter P-97 horizontal puller from thick-walled filamented boro-silicate glass (GC150F-7.5; Harvard Apparatus, UK) and fire-polished just before use with a microforge to a resistance of 8–12 MΩ. In order to minimize electrical noise, pipette tips were coated with Sylgard 184 (Dow Corning, Dow Silicones, UK). Cell-attached single-channel currents were recorded with an Axopatch 200B in patch configuration with a gain set to 500. The built-in low pass Bessel filter was set to 10 kHz and holding voltage was +100 mV. Single-channel currents were sampled at 100 kHz with a Digidata 1440 digitizer. For analysis, currents were additionally filtered with a 3 kHz low pass Gaussian filter and resampled to 33.3 kHz with Clampfit 10.7 software. In the recording pipette, agonists were dissolved in the pH 5 extracellular solution. Openings of GlyR in the presence of high agonist concentrations appear as clusters of openings, namely stretches of high open probability activity separated by long desensitized closed times that are not concentration dependent. We measured open probability with Clampfit 10.7 in clusters that had no double openings and were at least 100 ms long. Openings and closing events within the cluster were idealized with a threshold crossing method and cluster open probability was obtained as the ratio between the total open time and the duration of the cluster. Boxplots showing Popen values were created with OriginPro 2019 (OriginLab).

Nonparametric randomization test (two-tail, non-paired; 50,000 iterations) was used to determine p values for the difference between cluster open probabilities being greater than or equal to the observed difference (DC-Stats software: *Plested and Lape, 2020*).

## Thermodynamic cycle analysis

For each agonist, the maximum open probability reached, $maxP_{open}$, is linked to the equilibrium constant for gating by a simple relation.

$$maxP_{open} = \frac{E_{eff}}{E_{eff} + 1} \tag{1}$$

where $E_{eff}$ is the overall gating constant (e.g., incorporating the flipping/priming and the opening steps; *Burzomato et al., 2004*). We used *Equation 1* to estimate $E_{eff}$ from the $maxP_{open}$ values measured from clusters, that is, obtaining one $E_{eff}$ value per cluster.

The structures of four agonists used in this work were closely related and could be arranged in the cycle shown in *Figure 7*. Swapping the carboxylate groups of glycine and β-alanine with sulfonate

gives AMS and taurine, respectively. Similarly, lengthening the carbon chain of glycine and AMS with one methylene group gives β-alanine and taurine, respectively.

The existence of these structural relations between the agonists and our ability to estimate for each agonist a gating constant allowed us to carry out a thermodynamic cycle analysis (**Carter et al., 1984**; **Lee and Sine, 2005**) in order to assess whether the effect of combining the two structural modifications was predictable from the effects of each of the two modifications alone. For each structural modification of the agonist, the free energy change is given by:

$$\Delta G = -RTln\left(\frac{E_{eff\ m}}{E_{eff}}\right) \qquad (2)$$

where $E_{eff\ m}$ is the gating constant for the modified agonist, R is the gas constant (1.987 cal K$^{-1}$ mol$^{-1}$), and T is the absolute temperature (295K). The free energy change is the same along both pathways (see **Figure 7**).

$$\Delta G_1 + \Delta G_2' = \Delta G_2 + \Delta G_1' \qquad (3)$$

The coupling free energy $\Delta\Delta G_{int}$ of a thermodynamic cycle can be obtained as the difference between $\Delta G_1 - \Delta G_1'$ or $\Delta G_2 - \Delta G_2'$. If the $\Delta\Delta G_{int}$ is equal to 0, there is no interaction between the two structural changes, and their combined effect equals the sum of the individual effects.

We calculated the standard deviation of the mean for the free energy change and the free energy of coupling ($\Delta\Delta G_{int} = \Delta G_1 - \Delta G_1'$) from their distributions obtained by bootstrapping (10,000 runs with replacement) the $E_{eff}$ values from each agonist.

## Protein expression and purification

cDNA encoding the zebrafish full-length GlyR α1 (NP_571477) was cloned into the vector pFastBac1 for baculovirus expression in Sf9 cells (Thermo Fisher Scientific). Cells were tested routinely to check they were free from mycoplasma by DNA fluorochrome staining (CELLshipper Mycoplasma Detection Kit M-100 from Bionique).

A thrombin site and an 8× His tag were added at the C-terminus. Sf9 cells were cultured in SF9-900 III SFM at 27°C. To express GlyR, Sf9 cells, at the density of 2–3 million/ml, were infected by the baculovirus followed by incubation at 27°C for 72 hr.

Cells were harvested by a J20 centrifuge at 5000×*g* for 20 min. The cell pellet was suspended in 200 ml ice-cold buffer made of 20 mM Tris pH 8.0 and 150 mM NaCl (TBS), supplemented with 0.8 μM aprotinin, 2 μg/ml Leupeptin, 2 mM pepstain A, and 1 mM phenylmethylsulfonyl fluoride (PMSF). The cells were then disrupted by sonication. The large debris was removed by centrifugation at 10,000×*g* for 10 min. Membrane was pelleted by a Ti45 rotor at 45k rpm for 1 hr, resuspended in ice-cold TBS and homogenized by a Dounce homogenizer. The membrane was then mixed with a final concentration of 1% SMA copolymer XIRAN 30010 and incubated in a cold room for 2 hr. After centrifugation at 45k rpm for 1 hr using a Ti45 rotor, the supernatant was then incubated with 10 ml pre-equilibrated Ni-NTA beads for 6 hr. The Ni-NTA beads were loaded to an XK-16 column and washed by 10-column TBS containing 35 mM imidazole. The GlyR was eluted by 250 mM imidazole. Size exclusion chromatography (SEC) was then performed to further purify the protein.

## Cryo-EM sample preparation and data collection

The GlyR peak fraction eluted from SEC was diluted to 80 μg/ml. Considering the instability of AMS, a 100 mM AMS solution was prepared in TBS immediately before freezing the grids. Equal volumes of GlyR prep and 100 mM AMS were mixed, resulting in a final receptor concentration of 40 μg/ml. Dissolving AMS in TBS to a concentration of 100 mM produced a solution with pH 4.6. This rose to 4.95 when mixed with an equal volume of TBS (in order to estimate the final pH of the sample). At this pH, we expect 85% of AMS to be in zwitterion form.

Application of samples to glow-discharged Quantifoil 2/2 grids covered by 2 nm continuous carbon grids (3.5 μl per grid) was followed by flash-freezing in liquid ethane cooled by liquid nitrogen, using an FEI Mark IV cryo-plunge instrument.

The data set was collected on a Titan Krios operated at 300 kV, equipped with a BioQuantum K3 camera, using CDS mode at the magnification of 105,000×, corresponding to super-resolution pixel

size of 0.4155 Å. The defocus range was set from –1.2 to –2.2 μm. Each micrograph was recorded with a dose rate of 8 e⁻/Å²/s, resulting a total dose of 34.4 e⁻/Å².

## Image processing

The motion correction, CTF estimation, and particle picking were performed by cryoSparc (*Punjani et al., 2017*). The desensitized state glycine-bound GlyR in SMA (EMD-20388) was used as the initial model. After the extraction of particles, two rounds of 3D classification were performed by cryoSparc to remove the junk particles. One round of heterogeneous refinement with six classes was performed. Two classes with GlyR features containing 449,976 particles were selected. A second round of the heterogeneous refinement with four classes was performed and one class with good TMD features containing 283,117 particles was chosen. This was followed by a round of non-uniform refinement performed in cryoSparc. These good particles were then exported to Relion 3.1 (*Zivanov et al., 2018*) for one round of 3D classification. During the 3D classification in Relion 3.1, the particle alignment was closed and the data set was separated into eight classes with the T value of 20. The 3D classes with good TMD are selected for the subsequent 3D reconstruction. The final maps were generated by a direct reconstruction using Relion (*Zivanov et al., 2018*). The final FSC curves and local resolution maps were estimated by cryoSparc (*Punjani et al., 2017*). All maps are sharpened by LocScale (*Jakobi et al., 2017*).

## Model building

The model building commenced with the replacement of the ligand derived from the prior GlyR structures bound with taurine. The initial models for AMS-bound open, desensitized, and expanded-open states were taurine-bound open, desensitized, and expanded-open, respectively (*Yu et al., 2021*). The procedure to build each of the three AMS-bound states was the same. Take AMS-bound open state for example: the taurine-bound open state was first rigid-body fitted to the corresponding map using UCSF Chimera software (*Pettersen et al., 2004*). Coot (*Emsley and Cowtan, 2004*) was used to replace the ligand taurine with AMS. The structure was then manually adjusted in Coot followed by a round of Phenix (*Afonine et al., 2018*) refinement. The map to model cross-correlation values between the final model and map was 0.78, 0.84, and 0.81 for open, desensitized, and expanded-open states, respectively.

Note that our numbering of residues of the zebrafish GlyR α1 subunit includes the signal peptide which is predicted to be 16 residues.

## Acknowledgements

The authors would like to thank George Papageorgiou (The Francis Crick Institute, London, United Kingdom) for discussions on the stability of AMS.

---

# Additional information

### Funding

| Funder | Grant reference number | Author |
|---|---|---|
| Medical Research Council | Project grant MR/R009074/1 | Lucia G Sivilotti |
| National Institutes of Health | R01 GM100400 | Eric Gouaux |

The funders had no role in study design, data collection and interpretation, or the decision to submit the work for publication.

### Author contributions

Josip Ivica, Hongtao Zhu, Conceptualization, Formal analysis, Investigation, Methodology, Writing – original draft, Writing – review and editing; Remigijus Lape, Conceptualization, Formal analysis, Investigation, Writing – review and editing; Eric Gouaux, Conceptualization, Formal analysis, Supervision,

Funding acquisition, Investigation, Methodology, Writing – original draft, Project administration, Writing – review and editing; Lucia G Sivilotti, Conceptualization, Formal analysis, Supervision, Funding acquisition, Investigation, Writing – original draft, Project administration, Writing – review and editing

Author ORCIDs
Hongtao Zhu (iD) http://orcid.org/0000-0003-1522-0500
Eric Gouaux (iD) http://orcid.org/0000-0002-8549-2360
Lucia G Sivilotti (iD) http://orcid.org/0000-0003-2510-8424

Decision letter and Author response
Decision letter https://doi.org/10.7554/eLife.79148.sa1
Author response https://doi.org/10.7554/eLife.79148.sa2

## Additional files

### Supplementary files
• Supplementary file 1. Statistics for 3D reconstruction and model refinement.
• MDAR checklist

### Data availability
The coordinates and volumes for the cryo-EM data have been deposited in the Electron Microscopy Data Bank under accession codes EMD-26316, EMD-26315, and EMD-26317. The coordinates have been deposited in the Protein Data Bank under accession codes 7U2N, 7U2M and 7U2O. All data generated during this study is included in the manuscript and supporting files. Source data spreadsheets are provided for the electrophysiology data.

The following datasets were generated:

| Author(s) | Year | Dataset title | Dataset URL | Database and Identifier |
|---|---|---|---|---|
| Zhu H, Gouaux E | 2022 | A novel compound mimics the structural and functional effects of the full agonist glycine on glycine channels-desenstized state | https://www.emdataresource.org/EMD-26315 | EMDataResource, EMD-26315 |
| Zhu H, Gouaux E | 2022 | A novel compound mimics the structural and functional effects of the full agonist glycine on glycine channels-open state | https://www.emdataresource.org/EMD-26316 | EMDataResource, EMD-26316 |
| Zhu H, Gouaux E | 2022 | A novel compound mimics the structural and functional effects of the full agonist glycine on glycine channels-expanded open state | https://www.emdataresource.org/EMD-26317 | EMDataResource, EMD-26317 |
| Zhu H, Gouaux E | 2022 | A novel compound mimics the structural and functional effects of the full agonist glycine on glycine channels-desenstized state | https://www.rcsb.org/structure/7u2m | RCSB Protein Data Bank, 7U2M |
| Zhu H, Gouaux E | 2022 | A novel compound mimics the structural and functional effects of the full agonist glycine on glycine channels-open state | https://www.rcsb.org/structure/7u2n | RCSB Protein Data Bank, 7U2N |
| Zhu H, Gouaux E | 2022 | A novel compound mimics the structural and functional effects of the full agonist glycine on glycine channels-expanded open state | https://www.rcsb.org/structure/7u2o | RCSB Protein Data Bank, 7U2O |

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
