## [Editor Report]

Ivica et al. provide both functional and structural characterization of a relatively unstudied glycine receptor agonist. Their work supports their prior conclusions regarding the function of full vs. partial agonists, and provides a new look at a ligand that is structurally in between a full and partial agonist. This manuscript will be of interest to both biophysical and pharmacological investigations of ligand-gated ion channels.

---

## [Decision Letter]

**Decision letter after peer review:**

Thank you for submitting your article "Aminomethanesulfonic acid illuminates the boundary between full and partial agonists of the pentameric glycine receptor" for consideration by *eLife*. Your article has been reviewed by 2 peer reviewers, including Marcel P Goldschen-Ohm as the Reviewing Editor and Reviewer #1, and the evaluation has been overseen Kenton Swartz as the Senior Editor.

The reviewers have discussed their reviews with one another; please address all of the reviewer comments below.

*Reviewer #1 (Recommendations for the authors):*

1. lines 234-240, 290-292 – The differences in the binding site structures between glycine, AMS, and taurine in Figure 5D, E appear to be incredibly small with respect to the resolution of the structural models. I am skeptical about all of the conclusions that the authors draw in this section about taurine not producing as compact a binding site as glycine or AMS and glycine shifting loops B and C closer to the TMD. Based on the model resolutions, wouldn't it be more apt to conclude that there are no obvious differences in binding site compactness, etc.? If the authors disagree, could you please offer some additional explanation as to why the very small differences (which the authors refer to as "subtle") should be considered sufficiently well resolved to be meaningful? Perhaps densities can be used to make an argument here? Also, the same question for Figure 6 ECD-TMD interface, regarding subtle differences in compactness of this interface with taurine vs glycine or AMS bound.

2. lines 177, 237, 251 – related to above, please define what "subtle" means.

3. The relative lack of a desensitized conformation in the SMA-extracted glycine receptor particles as compared to what would be expected based on functional data should be discussed. Part of this discussion could simply reference the comparison between nanodiscs and SMA in Yu et al., 2021, but it should be at least briefly discussed here as it is highly relevant to the interpretation of the structures.

4. In Figure 3—figure supplement 2, it is clear that the open, desensitized, and expanded-open clusters were selected based on their similarity to previously identified structures. However, the desensitized and expanded-open clusters account for only a couple percent of the total particles, whereas other clusters that were discarded account for larger fractions of the particles. Could the authors please provide a more detailed discussion of what criteria were used to discard these other clusters?

5. line 342 – extra "that".

*Reviewer #2 (Recommendations for the authors):*

1. The authors write (lines 60-62) that the binding of high efficacy agonists like glycine leads to channels adopting either open or desensitized states, while binding of partial agonists results in an additional third (intermediate) state that represents agonist-bound closed conformations of the channel. This isn't really true, however, or is an oversimplification that could be confusing to the reader. If a maximally-effective concentration of glycine has a Po of 0.96, there clearly are brief transitions to glycine-bound closed states of the channel. Wouldn't it be more accurate to state that as agonist efficacy increases, the duration of ligand-bound closed states decreases? Also, what about when low (non-saturating) concentrations of glycine are used? Would one still expect the two states the authors mention?

2. In Figure 4D, the AMS_Open and Taurine_Open lines look more similar to one another (except when the distance along the pore axis >30A) than they do with the Glycine_Open line. Also, the taurine and AMS open lines both look markedly different from the glycine line between 10 and 18 angstroms along the pore axis. What is the significance of this?

3. The single channel data obtained using 100 mM taurine assumes that this concentration is truly maximally-effective. Is this the case? If it is not, could that explain the statement on line 336 that "…taurine is less efficacious than we would predict."?

4. Line 320. How do max Po values of 0.96 and 0.54 yield Eeff values of 60 and 3.8 from the equation Po = Eeff/(1 + Eeff)? Shouldn't they be closer to 24 and 1.2? The same question applies to AMS and taurine on line 321.

5. Do the authors have any thoughts on why the Po, and thus efficacy, of β-alanine and especially taurine went up relative to glycine as the pH was raised from 5 to 7.4 (Table 2)?

---

## [Author Response]

Reviewer #1 (Recommendations for the authors):1. lines 234-240, 290-292 – The differences in the binding site structures between glycine, AMS, and taurine in Figure 5D, E appear to be incredibly small with respect to the resolution of the structural models. I am skeptical about all of the conclusions that the authors draw in this section about taurine not producing as compact a binding site as glycine or AMS and glycine shifting loops B and C closer to the TMD. Based on the model resolutions, wouldn't it be more apt to conclude that there are no obvious differences in binding site compactness, etc.? If the authors disagree, could you please offer some additional explanation as to why the very small differences (which the authors refer to as "subtle") should be considered sufficiently well resolved to be meaningful? Perhaps densities can be used to make an argument here? Also, the same question for Figure 6 ECD-TMD interface, regarding subtle differences in compactness of this interface with taurine vs glycine or AMS bound.

We appreciate the comments from this reviewer. We agree that the conformation of the binding sites showing in Figure 5E is similar for glycine, AMS, and taurine. Considering the conformational changes of the binding pockets involves a collection of multiple residues, the binding pocket volume can thus more clearly reflect differences in conformations between the different agonists. The volumes of the glycine, AMS and taurine binding pockets are 130, 125 and 151 Å^3^ respectively. These measurements show that the taurine binding pockets are larger than glycine and AMS binding pockets, consistent with the notion that taurine is not producing as compact a binding pocket as glycine and AMS. The text in Lines 239 and following is revised to read “The volume of glycine, AMS and taurine binding pockets are 130, 125 and 151 Å^3^ respectively, showing that the binding pocket, when bound with AMS or glycine, takes up a conformation that is more compact than that seen with the partial agonist taurine”.

For the ECD-TMD interface, we agree that the measurements for ECD-TMD interface are similar. We revised the text in Lines 267 and following to read “We found that the ECD-TMD interface are overall similar between glycine, AMS and taurine (Figure 6A-B), as indicated by the distances between the centers of mass of the secondary structure elements.”

2. lines 177, 237, 251 – related to above, please define what "subtle" means.

We agreed with the reviewer’s comments above. Related with Line 177 (old line numbering), we have revised the text in Line 177 and following to read “Importantly, these reconstructions have well-resolved extracellular domain (ECD) and transmembrane domain (TMD) densities, allowing us to observe conformational differences (Figure3—figure supplement 3, Extended-Table 1) in the structures.”

Related with Lines 237 and 251, please see our replies above.

3. The relative lack of a desensitized conformation in the SMA-extracted glycine receptor particles as compared to what would be expected based on functional data should be discussed. Part of this discussion could simply reference the comparison between nanodiscs and SMA in Yu et al., 2021, but it should be at least briefly discussed here as it is highly relevant to the interpretation of the structures.

We appreciate these comments. The related discussion was added with the reference to Yu et al., 2021 cited in Lines 304 and following to read “Indeed, our data shows that, for the AMS-bound receptor, 94.3 % and 3.4% of the particles are in open and desensitized state classes, respectively (Extended-Table 1), consistent with the notion that the particle fractions for desensitized states are much smaller than open states when the receptor is isolated via SMA compared to when it is incorporated into nanodiscs (Yu et al. 2021).”

4. In Figure 3—figure supplement 2, it is clear that the open, desensitized, and expanded-open clusters were selected based on their similarity to previously identified structures. However, the desensitized and expanded-open clusters account for only a couple percent of the total particles, whereas other clusters that were discarded account for larger fractions of the particles. Could the authors please provide a more detailed discussion of what criteria were used to discard these other clusters?

We appreciate these comments. The criteria by which we discarded the ‘bad’ 3D classes generated in Relion were based on TMD features. As shown in Author response image 1, the classes with weak and disconnected TMD were discarded. More details about the 3D classification are added in the Image processing part of the methods section to read “After the extraction of particles, two rounds of 3D classification were performed by cryoSparc to remove the junk particles. One round of heterogeneous refinement with 6 classes was performed. Two classes with GlyR features containing 449,976 particles were selected. A second round of the heterogeneous refinement with 4 classes was performed and one class with good TMD features containing 283,117 particles was chosen. This was followed by a round of non-uniform refinement performed in cryoSparc. These good particles were then exported to Relion 3.1 (Zivanov et al. 2018) for one round of 3D classification. During the 3D classification in Relion 3.1, the particle alignment was closed and the data set was separated into 8 classes with the T value of 20. The 3D classes with good TMD are selected for the subsequent 3D reconstruction.”

**Author response image 1. sa2fig1:** 

5. line 342 – extra "that".

Done.

Reviewer #2 (Recommendations for the authors):1. The authors write (lines 60-62) that the binding of high efficacy agonists like glycine leads to channels adopting either open or desensitized states, while binding of partial agonists results in an additional third (intermediate) state that represents agonist-bound closed conformations of the channel. This isn't really true, however, or is an oversimplification that could be confusing to the reader.

The wording of those lines was imprecise, mixing structure and function. This has now been rewritten to be more accurate (lines 51 and following)

If a maximally-effective concentration of glycine has a Po of 0.96, there clearly are brief transitions to glycine-bound closed states of the channel. Wouldn't it be more accurate to state that as agonist efficacy increases, the duration of ligand-bound closed states decreases? Also, what about when low (non-saturating) concentrations of glycine are used? Would one still expect the two states the authors mention?

When the receptor is saturated by the agonist, the proportion of time spent in closed ligand-bound states decreases as the efficacy of the agonist increases.

We do not have structural data at low, non- saturating concentrations of glycine, but we have done the calculations with the best functional model we have for glycine vs. taurine (Lape et al., 2008). Author response table 1 shows the difference in equilibrium occupancy of the states we would see in clusters for three conditions – saturating glycine, saturating taurine and low glycine concentration (chosen to produce the same Popen – 56% -as saturating taurine).

**Author response table 1. sa2table1:** 

EQUILIBRIUM	OCCUPANCY					
		0.08mM gly	100 mM taurine	10 mM gly		
Open (=Popen)		0.56		0.56		0.95
						
bound closed	0.18		0.44		0.05	
						
unbound closed	0.26		0.00		0.00	

The calculations show that at low glycine, more than 50% of the shut time is due to the unliganded state (which is effectively absent in the presence of high agonist concentrations), but there is substantial occupancy of the bound closed states, although this is lower than the level predicted for high taurine. It is therefore possible that the additional shut state seen with partial agonists would be structurally detectable at low glycine, but the relation between structural and functional data is only semi quantitative, and we’d rather keep these considerations for further work, as they are too speculative for this paper.

2. In Figure 4D, the AMS_Open and Taurine_Open lines look more similar to one another (except when the distance along the pore axis >30A) than they do with the Glycine_Open line. Also, the taurine and AMS open lines both look markedly different from the glycine line between 10 and 18 angstroms along the pore axis. What is the significance of this?

We agree with the reviewer that AMS_Open and Taurine_Open lines are similar to each other, underscoring the notion that nearly full and partial agonists result in opening of the ion channel to similar open conformations.

We also agree that the Glycine_Open line looks different between 10 and 18 Å. The differences occur at the 9’L (L277), which functions as the constriction point in the Apo and partial agonist bound closed states (Yu et al., 2021, Cell). Upon activation, the side chain of 9’L rotates and the channel opens. While the density of the 9’L becomes weaker at the end of the side chain, the density for the Cα and Cβ are strong and, when we lower the contour level, we can visualize the entire side chain. As a result, we believe that the ion channel radii, as defined by the position of the 9’L side chain, are different. In Author response image 2, we show models and maps, where the maps are all contoured at 4.9σ. While the overall open state conformations of the ion channel are similar between AMS, taurine and glycine, in the glycine bound state there are subtle differences in the conformation of the open state, most notably at 9’L. At present, however, we do not fully understand the molecular mechanism underpinning this difference and await further studies, such as long timescale molecular dynamics simulations.

3. The single channel data obtained using 100 mM taurine assumes that this concentration is truly maximally-effective. Is this the case? If it is not, could that explain the statement on line 336 that "…taurine is less efficacious than we would predict."?

The single channel data were obtained using 500 mM taurine, not 100 mM. In the whole-cell dose-response experiments shown in Figure 1 , even 300 mM taurine appears to be maximally-effective. We chose 500 mM for the single channel recording to err on the side of caution- as experimental time was limited. Given that these agonists do not block the channel (contrast with nicotinic acetylcholine receptors), using a supramaximal concentration was not a concern.

4. Line 320. How do max Po values of 0.96 and 0.54 yield Eeff values of 60 and 3.8 from the equation Po = Eeff/(1 + Eeff)? Shouldn't they be closer to 24 and 1.2? The same question applies to AMS and taurine on line 321.

In the original manuscript, the Eeff values were calculated from the Popen values as follows: each cluster (longer than 100 ms and with no double openings) gave one Popen and one Eeff value. In the text we then quoted the mean Eeff values for the different agonists.

However, the relation of Eeff to Popen is non-linear, and neither the Popen nor the Eeff values are normally distributed, and this causes the discrepancy spotted by the referee. In our original analysis we had taken into account by reporting both mean and median values of Popen (Table 2) and by using non-parametric statistical tests. However, we had cited mean Eeff values in the text and in the calculations. We have revised this to use the median Popen to calculate Eeff and the energy values in the cycle– the conclusions of the paper are not changed.

5. Do the authors have any thoughts on why the Po, and thus efficacy, of β-alanine and especially taurine went up relative to glycine as the pH was raised from 5 to 7.4 (Table 2)?

Much of this difference is expected because of the non-linear relation of Popen to Eeff, eg. Popen = Eeff/(Eeff+1) and the differences in the initial values of efficacy for the three agonists.

Author response table 2 shows what we expect to happen if we have a general enhancement of gating and an identical increase in efficacy for all our three agonists.

**Author response table 2. sa2table2:** 

	pH 5	Effect of gating enhancement	Popen	5x Eeff
Median Popen	Eeff
Glycine	**0.976**	40.7	0.995	203.5
Β-alanine	**0.565**	1.3	0.867	6.5
Taurine	**0.057**	0.06	0.231	0.3

The first two columns show our measurements of maximum Popen (bold) and the estimates of efficacy we obtained at pH 5, the second pair of columns shows the effect of increasing efficacy equally for all agonists by five-fold at pH 7.4.

The increase in Popen for glycine is so small (from 0.976 to 0.995) that it would not be detectable, but the Popen increases for both of the less efficacious agonists, β-alanine and taurine, are large and quite obvious.

As the reviewer noted, at pH 7.4 the Popen of taurine may be higher than what we would expect if the efficacy of all agonists increased by the same amount. It is hard to be sure that this is a real difference given the large scatter of the data for this agonist. The relevant data at pH 7.4 are shown in Figure 4 4C in Ivica et al., 2021 (doi: 10.1074/jbc.RA119.012358)